# TRANSFORMERS CAN DO BAYESIAN INFERENCE

**Samuel Müller[1], Noah Hollmann[2], Sebastian Pineda[1], Josif Grabocka[1], Frank Hutter[1,3]**
[1]University of Freiburg, [2]Charité Berlin, [3]Bosch Center for Artificial Intelligence
Correspondence to Samuel Müller: `muellesa@cs.uni-freiburg.de`

## ABSTRACT

Currently, it is hard to reap the benefits of deep learning for Bayesian methods, which allow the explicit specification of prior knowledge and accurately capture model uncertainty. We present *Prior-Data Fitted Networks (PFNs)*. PFNs leverage large-scale machine learning techniques to approximate a large set of posteriors. The only requirement for PFNs to work is the ability to sample from a prior distribution over supervised learning tasks (or functions). Our method restates the objective of posterior approximation as a supervised classification problem with a set-valued input: it repeatedly draws a task (or function) from the prior, draws a set of data points and their labels from it, masks one of the labels and learns to make probabilistic predictions for it based on the set-valued input of the rest of the data points. Presented with a set of samples from a new supervised learning task as input, PFNs make probabilistic predictions for arbitrary other data points in a single forward propagation, having learned to approximate Bayesian inference. We demonstrate that PFNs can near-perfectly mimic Gaussian processes and also enable efficient Bayesian inference for intractable problems, with over 200-fold speedups in multiple setups compared to current methods. We obtain strong results in very diverse areas such as Gaussian process regression, Bayesian neural networks, classification for small tabular data sets, and few-shot image classification, demonstrating the generality of PFNs. Code and trained PFNs are released at `https://github.com/automl/TransformersCanDoBayesianInference`.

## 1 INTRODUCTION

In the last decade, supervised machine learning (ML) methods using deep learning architectures have made substantial progress on machine learning tasks where a large amount of training data is available (Vaswani et al., 2017; He et al., 2016; Krizhevsky et al., 2012). A very important problem in ML is thus to transfer these successes to smaller-scale setups with less data available. In this paper, we propose a way to build models that approximate posteriors with flexible and replaceable priors using deep learning models. It makes specifying a prior as simple as defining a sampling scheme of supervised learning tasks.

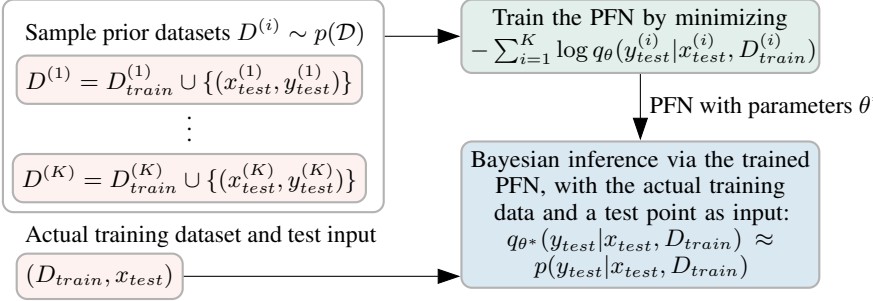

Figure 1: A visualization of Prior-Data Fitted Networks (PFNs). We sample datasets from a prior and fit a PFN on hold-out examples of these datasets. Given an actual dataset, we feed it and a test point to the PFN and obtain an approximation to Bayesian inference in a single forward propagation.

While the success of deep learning on large datasets can be attributed to the capacity of neural networks to approximate any function, there is still a need for encoding prior knowledge, for example through model architecture (e.g., Convolutional Neural Networks (LeCun et al., 1989)) or regularizers (e.g., data augmentations (Hendrycks et al., 2019; Cubuk et al., 2020)). Otherwise, the no free lunch theorems show that there are no good methods to solve the class of prediction problems (Wolpert & Macready, 1997). Thus, a large number of specialized algorithms have been developed for different small-scale tasks (LeCun et al., 1989; Kadra et al., 2021; Chen & Guestrin, 2016). Encoding prior information into a machine learning model can, however, be challenging.

A well-defined way to bias a model is to use Bayesian inference. The foundation of Bayesian inference is an assumption on the distribution of the data to appear in a real-world application. This assumption gives rise to a prior belief over the probability for the data to follow a particular model. One might, e.g., implement a prior, encoding the data as created by a neural network (Bayesian Neural Networks, (MacKay, 1992)), by a polynomial, the likelihood of a Gaussian mixture or random code in a pre-defined programming language (Solomonoff, 1997). Using Bayesian inference for prediction in supervised learning has the advantages that (i) it has a theoretical foundation that makes it valid in setups where the prior $p(t)$ fits; (ii) it can thus better account for the actual likelihood of different events; (iii) it is well calibrated and (iv) it is interpretable as the prior describes the expectations of the model. However, retrieving the posterior predictive distribution for a given prior is intractable in most cases (Blei et al., 2017; MacKay, 1992).

Figure 1 outlines Prior-Data Fitted Networks (PFNs), for approximating such Bayesian models. We assume a given representative prior distribution over supervised learning tasks (or functions), which provides our inductive bias. To train a PFN, we use supervised learning with set-valued inputs representing entire datasets: we repeatedly sample a meta-train task (or function) from the given prior, draw a set of data points and their labels from it, mask one of the labels and learn to make probabilistic predictions for it based on the set-valued input of the rest of the data points. Given an actual real-world dataset, we feed it together with a test point as inputs to the PFN and that outputs its prediction distribution for the test point, conditioned on the dataset. As we will demonstrate, this distribution approximates exact Bayesian posterior prediction. We refer to this step as (Bayesian) inference, as opposed to the training of the PFN itself.

Our PFNs thus allow us to approximate the posterior predictive distribution for *any* prior from which we are able to sample data. This is a very weak requirement compared to the standard assumptions of other approximations for Bayesian inference (Hoffman et al., 2014; 2013; Jordan et al., 1999). This allows a simple approximation of a large set of priors, including priors that are very hard to approximate with currently available tools. We make the following contributions:

- We present architectural changes to successfully use Transformers for posterior predictive distribution (PPD) approximation, including a novel predictive distribution for regression tasks. The proposed method is simple, cheap and generally applicable to a large set of priors.

- We demonstrate that PFNs can approximate the PPD of Gaussian processes and Bayesian neural networks (BNNs) orders of magnitude faster than MCMC with NUTS or SVI with Bayes-by-Backprop (Blundell et al., 2015).

- We demonstrate that PFNs can have an impact on real-world tasks. (i) We implement BNNs with priors over architectures on PFNs that enable tuning free predictions in a single forward pass and outperform all baselines on a large benchmark of small tabular datasets. (ii) Additionally, we show that simple handwriting priors enable few-shot learning on Omniglot (Lake et al., 2015).

## 2 BACKGROUND

This paper concerns the use of a large amount of data to later yield strong performance on tasks for which only small datasets are available. In multiple machine learning setups, one can see successes based on this approach, such as fine-tuning unsupervised models on image data (Chen & He, 2021; Chen et al., 2020) or large language corpora (Devlin et al., 2018). Another way to transfer knowledge from large corpora is to design prompts for language models (Wang & Komatsuzaki, 2021; Brown et al., 2020). While we use a large amount of data during training, unlike the other approaches, all the data we use is generated artificially.

In meta-learning, or learning-to-learn (Hochreiter et al., 2001; Finn et al., 2017a), one tries to generalize learning methods from a training set of datasets to a validation set of datasets. Similar to our approach, there is a branch of research on building models that can learn to learn in a single forward pass (Santoro et al., 2016; Hochreiter et al., 2001; Gordon et al., 2019). More recently (Conditional) Neural Processes (Garnelo et al., 2018a;b; Kim et al., 2018) were proposed for meta-learning. These methods focus on architectural constructions as stochastic processes. The difference, besides the dissimilar model architectures and dissimilar prediction distributions, to previous meta-learning work is that we set up learning problems from which we sample artificial, on-the-fly-generated datasets such that we can show that our method does not only learn to learn, but that it learns to perform Bayesian inference.

In this work, we are interested in Bayesian posterior prediction for supervised learning problems, i.e., we model the output $y$ for a new input $x$ based on a supervised dataset of arbitrary size $n$, $\mathcal{D} = \{(x_i, y_i)\}_{i=1}^n$, where $y_i$ is the *output* for $x_i$. To address the supervised learning problem in a Bayesian way, we consider a prior $p(t)$ over the latent variable $t$, the task. We can now use Bayes' Theorem to define the posterior $p(t|\mathcal{D})$. The crucial distribution for prediction, the posterior predictive distribution (PPD), can then be inferred as

$$p(y|x, \mathcal{D}) = \int_t p(y|x, t)p(t|\mathcal{D}). \tag{1}$$

In some cases, one can arrive at the PPD $p(y|x, \mathcal{D})$ in closed form (e.g., for Gaussian Processes, (Rasmussen & Williams, 2005)), but in most cases the PPD can only be approximated. Traditionally, one approximates the posterior $p(t|\mathcal{D})$ and based on this approximation infers an approximation of the PPD. To approximate the posterior, there are two classes of prominent methods: i) Markov Chain Monte Carlo (MCMC) (Neal, 1996; Andrieu et al., 2003; Welling & Teh, 2011) methods, like NUTS (Hoffman et al., 2014), allow accurate but sometimes very slow approximations of the posterior; and ii) Variational Inference (VI) (Jordan et al., 1999; Wainwright & Jordan, 2008; Hoffman et al., 2013) methods approximate the posterior by a tractable distribution, such as a factorized normal distribution.

One more line of research to be mentioned is amortized simulation-based inference (Cranmer et al., 2020; Chan et al., 2018), in particular neural posterior estimation (Lueckmann et al., 2021; Papamakarios & Murray, 2016). The objective here is to find a model that approximates the posterior $p(t|X)$ by training on samples from $p(t, X)$, where $X$ can be a dataset $\mathcal{D}$ like above, but typically is a single vector. Previous work in simulation-based inference is focused on simulations for which a specific model underlying the data is known (Lueckmann et al., 2021). Our method is in so far similar to simulation-based inference as both solely use samples from the prior. In the next section, we will show how we can model the PPD directly, though, without ever instantiating the posterior, and apply this to large datasets with general priors.

## 3  PPD Approximation with PFNs

Let us consider a parameterized model $q_\theta$ that can accept a dataset $D = \{(x_i, y_i)\}_{i=1}^n$, as well as a query $x$ as input, and which predicts a distribution over possible values of $y$ for the query $x$. Many neural architectures can be used as such a model; in this paper, we use a variant of Transformers (Vaswani et al., 2017) as they are a reliable and powerful architecture. We train this model by

---

**Algorithm 1:** Training a PFN model by Fitting Prior-Data

**Input**  : A prior distribution over datasets $p(\mathcal{D})$, from which samples can be drawn and the number of samples $K$ to draw
**Output** : A model $q_\theta$ that will approximate the PPD
Initialize the neural network $q_\theta$;
**for** $j \leftarrow 1$ **to** $K$ **do**
    Sample $D \cup \{(x_i, y_i)\}_{i=1}^m \sim p(\mathcal{D})$;
    Compute stochastic loss approximation $\bar{\ell}_\theta = \sum_{i=1}^m (-\log q_\theta(y_i|x_i, D))$;
    Update parameters $\theta$ with stochastic gradient descent on $\nabla_\theta \bar{\ell}_\theta$;
**end**

---

cross-entropy over samples drawn from the prior. Our proposed loss, the *Prior-Data Negative Log-Likelihood (Prior-Data NLL)* $\ell_\theta$ is defined as

$$\ell_\theta = \mathbb{E}_{D \cup \{x,y\} \sim p(\mathcal{D})}[-\log q_\theta(y|x, D)], \tag{2}$$

where $D \cup \{x, y\}$ simply is a dataset of size $|D| + 1$ sampled from $p(\mathcal{D})$.

PFNs are trained to minimize $\ell_\theta$ like outlined in Algorithm 1: We draw many datasets from our prior and fit our model to predict a hold-out example correctly for these.

In contrast to VI and MCMC, we learn to approximate the PPD directly from dataset samples. For all cases we consider, it is simple and cheap to sample from $p(\mathcal{D})$, fulfilling the only requirement our method has. MCMC on the other hand requires access to a non-normalized posterior $f(t|D, x) \propto p(t|D, x)$, and VI requires access to the density values of $p(t, D)$, both of which can be hard to compute.

Following similar insights as Gordon et al. (2019), we can show that minimizing the Prior-Data NLL $\ell_\theta$ yields an approximation of the PPD in terms of cross-entropy and thus *KL*-Divergence:

**Insight 1.** *The proposed objective $\ell_\theta$ is equal to the expectation of the cross-entropy between the PPD and its approximation $q_\theta$: $\ell_\theta = \mathbb{E}_{x,D \sim p(\mathcal{D})}[H(p(\cdot|x, D), q_\theta(\cdot|x, D))]$*

*Proof.* The above can be shown with a simple derivation. We mark transformations for easy reading.

$$\ell_\theta = -\int_{D,x,y} p(x, y, D) \log q_\theta(y|x, D) = -\int_{D,x} p(x, D) \int_y p(y|x, D) \log q_\theta(y|x, D) \tag{3}$$

$$= \int_{D,x} p(x, D) \mathrm{H}(p(\cdot|x, D), q_\theta(\cdot|x, D)) = \mathbb{E}_{x,D \sim p(\mathcal{D})}[\mathrm{H}(p(\cdot|x, D), q_\theta(\cdot|x, D))] \tag{4}$$

$\square$

Thus, we have an approximation of the PPD. We can formulate this in terms of *KL*-Divergence, too.

**Corollary 1.1.** *The loss $\ell_\theta$ equals the expected KL-Divergence $\mathbb{E}_{D,x}[KL(p(\cdot|x, D), q_\theta(\cdot|x, D))]$ between $p(\cdot|x, D)$ and $q_\theta(\cdot|x, D)$ over prior data $x, D$, up to an additive constant. Proof in App. A.*

In the following, we consider the optimum of the given optimization problem $\theta^* \in \arg\min_\theta \ell_\theta$.

**Corollary 1.2.** *If $p$ is a member of the distribution family $q_\theta$ (i.e., there is a $\theta$ such that $q_\theta = p$), we have $q_{\theta^*}(\cdot|x, D) = p(\cdot|x, D)$ for all $x, D$ for which $p(\cdot|x, D)$ is defined. Proof in App. B.*

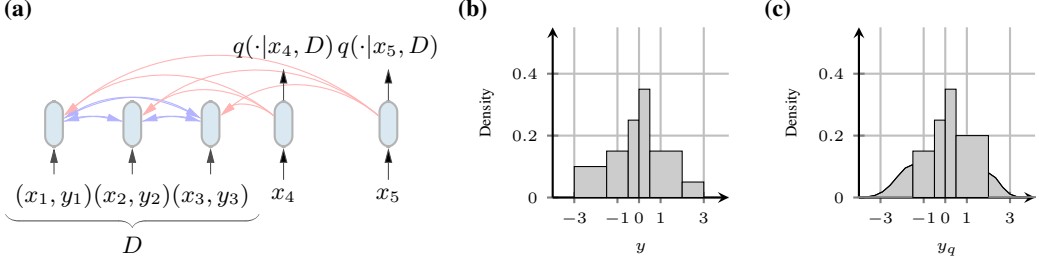

Figure 2: (a) A visualization of the Transformer for $n = 3$ input pairs and $m = 2$ queries. Every bar is the representation of one input and the arrows show what each representation can attend to. (b-c) A visualisation of the Riemann distribution, with and without bounded support.

## 4 ADAPTING THE TRANSFORMER FOR BAYESIAN INFERENCE

In this section, we propose an efficient architecture based on the Transformer encoder (Vaswani et al., 2017) and a novel regression head for regression problems.

**An Efficient Architecture** Our architecture, visualized in Figure 2a, is based on a Transformer encoder without positional encodings, which makes it invariant to permutations in the dataset $D$. We

feed the Transformer with simple linear projections of our inputs and queries. Our model returns the PPD for each given query $x$, only depending on the dataset $D$ and the query itself. We sample the number of inputs $n$ and queries $m$ randomly such that $N = m + n$ for some fixed $N$. See Appendix E.1 for a more detailed description and Appendix E.3 for an ablation of the permutation invariance. For additional practical training information see Appendix D.

**Riemann Distribution** The modelling of continuous distributions is hard with neural networks. To yield strong performance in modelling PPDs, we used a distribution that works particularly well with neural networks. Based on the knowledge that neural networks work well for classification and inspired by discretizations in distributional reinforcement learning (Bellemare et al., 2017), we made use of a discretized continuous distribution that we call *Riemann Distribution*. PDFs of this distribution are bar plots, see Figure 2b. We select the boundaries of the buckets $\mathbf{B}$ in which we discretize such that they all have the same probability in prior-data: $p(y \in b) = 1/|\mathbf{B}|, \forall b \in \mathbf{B}$. We estimate this using a large prior-data sample. In experiments with priors that need unbounded support, we replace the last bar on each side with an appropriately scaled half-normal distribution, see Figure 2c. For a more exact definition, we refer the reader to Appendix E.2. For an ablation of the Riemann distribution, see Appendix E.3.

**Theorem 1.** *A Riemann distribution $q_{\mathbf{B}}$ with finite support, where $\mathbf{B}$ are the buckets, can model any distribution with a finite, almost everywhere continuous (Riemann-integrable) density function and full support up to arbitrary precision. That is, $\forall p \forall \epsilon \exists \mathbf{B}.KL(q_{\mathbf{B}}, p) \leq \epsilon$. Proof in App. C.*

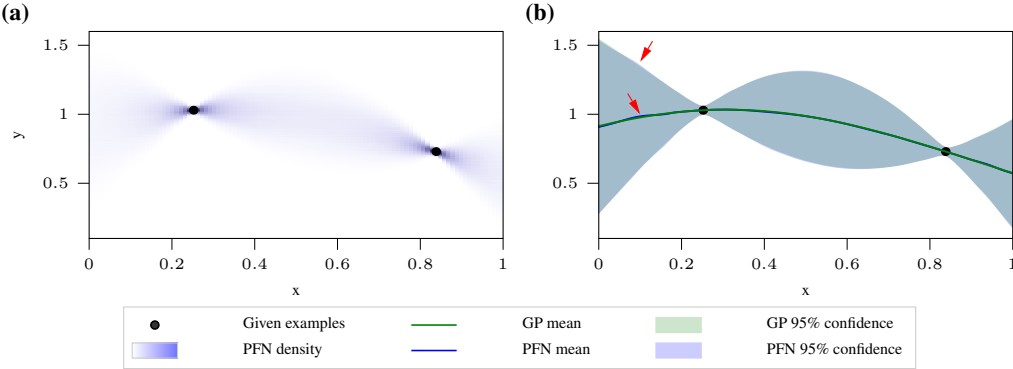

Figure 3: (a) The PFN's PPD given two evaluations of the function, highlighting the binning it its predictions (best viewed in full-screen mode). (b) The PFN's and the GP's mean and confidence intervals, which are nearly identical (red arrows mark tiny exemplary differences). For additional comparisons, see Figure 7 in the appendix.

## 5 POSTERIOR APPROXIMATION STUDIES

In our first set of experiments, we study the capability of PFNs to perform Bayesian inference for the tractable case of Gaussian Processes (GPs) with fixed hyperparameters (where we can compare to ground truth data; Section 5.1) and the intractable cases of GPs with unknown hyperparameters (Section 5.2) and Bayesian Neural Networks (BNNs; Section 5.3).

Corollary 1.1 shows that the Prior-Data NLL is equal to the *KL*-Divergence between the exact PPD and the approximation, up to an additive constant. We thus use it in all following experiments to compare how close methods can match the correct PPD. For methods that do not approximate the PPD directly, we approximate the integral of the PPD in Equation 1 with a large Monte Carlo integration. In this section, we report the 95% confidence interval over different sampled datasets of the Prior-Data NLL for each experiment as shaded areas alongside the mean. For more information of the setup of the following setup we refer the reader to Appendix F.

### 5.1 GAUSSIAN PROCESS APPROXIMATION

We begin by approximating GPs to study what PFNs are capable of; GPs with fixed hyperparameters are convenient for this purpose since the tractability of their PPD allows us to assess how close

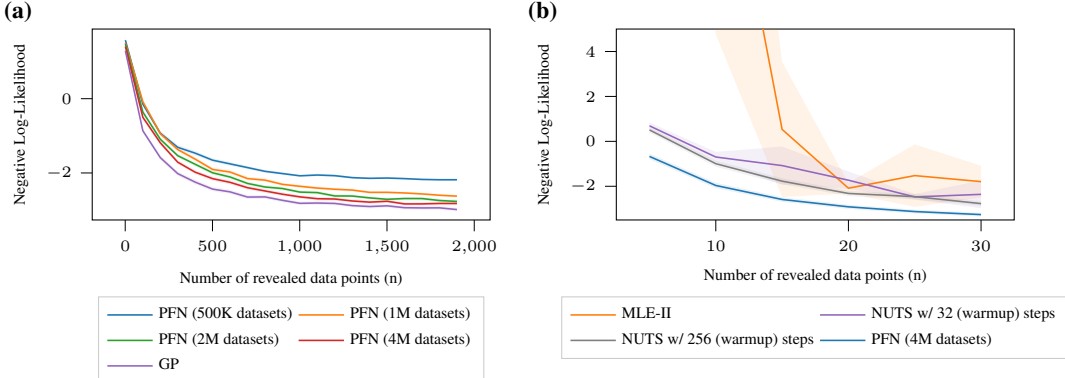

Figure 4: (a) Prior-Data NLL with a fixed GP. While, the PPD of the GP is exact, we can see that our approximations get very close to it. (b) Prior-Data NLL with a mix over GP hyperparameters. There is no closed-form solution for this PPD, but we can see that our PFN still gets closest to the true PPD. Our PFN is also more than $200\times$ faster than any of the baselines.

our approximation is to the exact PPD. To create the prior for this task, we sample the $N$ inputs $x_i$ uniformly at random from the unit-cube. The data-prior of a zero-mean GP with a kernel function $k$, given the $x_i$'s, can be described as a multivariate normal distribution $\mathcal{N}(\mathbf{0}, K)$, where $K_{i,j} = k(x_i, x_j)$. We sample from this normal distribution to yield a dataset $\{(x_1, y_1), \ldots, (x_N, y_N)\}$. As described above, we then train our PFNs to predict the $y$'s for a held-out subset of the dataset. The trained PFN can then be used to approximate the PPD for new datasets and arbitrary query points in the unit-cube.

Figure 3 visualizes the predictions of such a PFN with two data points and query points $x$ ranging from 0 to 1. We visualize both the estimated PPD as a heat map, as well as confidence bounds, compared to the exact PPD. The estimated confidence intervals and means are virtually indistinguishable from the ground truth, therefore we marked the most noticeable differences with red arrows. The model learns on its own, completely automatically, to generate smooth distributions without any explicit knowledge of the positions of the bars in $\mathbb{R}$. We published[1] an interactive version of this experiment. In Figure 7 of the appendix, we show more qualitative examples of the behavior of our approximation together with the exact posterior for a different length scale. In Figure 8, we additionally compare to Attentive Neural Processes (Kim et al., 2018), showing the clear superiority of our method for Bayesian Inference. Figure 4a shows that the qualitative results above generalize to datasets with up to 2000 examples with multiple features. The tractable GP posterior, which achieves optimal performance by definition, is approximated very closely, and increasingly better so the more meta-train datasets we train the PFN for.

## 5.2 APPROXIMATING A GP WITH HYPER-PRIORS

A common practice in fields applying GPs in the real world is to define distributions over the hyperparameters of these GPs (Rasmussen & Williams, 2006; Snoek et al., 2012), so-called hyperpriors. Thus, the prior of these models considers a more diverse set of functions, to model the fact that the correlation between data points, the smoothness of functions and the scale of outputs varies across applications. The downside of using hyper-priors is that it is not possible to compute the PPD of the GP exactly anymore. There are two common practices to approximate it anyways, and we evaluate against both. (i) Firstly, MLE-II, the most common approximation. This is a simple special case of variational inference. Here, suitable hyperparameters are found by maximum a posteriori (MAP) estimation (e.g., Rasmussen & Williams (2006)). We use the fitting setup of BoTorch (Balandat et al., 2020) which also inspired our hyper-priors. (ii) Secondly, Markov Chain Monte Carlo (MCMC), a method to sample from any distribution for which one can calculate non-normalized probabilities, that is also frequently applied to sample hyperparameters (e.g., Snoek et al. (2012)). Here, we use the state-of-the-art algorithm NUTS (Hoffman et al., 2014), which uses gradients to facilitate the

---

[1]https://huggingface.co/spaces/samuelinferences/\transformers-can-do-bayesian-inference

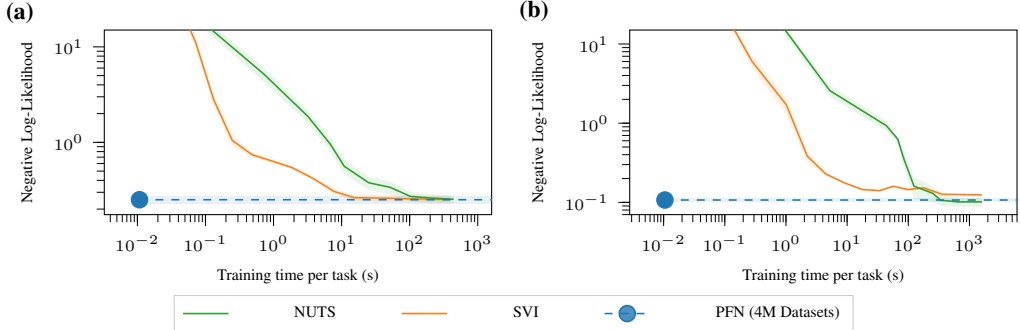

Figure 5: Time spent for Bayesian Inference per dataset compared to Prior-Data NLL. For NUTS we scale the number of warmup and sampling steps, for SVI, we scale the number of training steps and averaged samples. PFN is evaluated at only one setting (blue dot). We show two different BNN architectures (a) a BNN with 3 features, 2 layers and a hidden dimensionality of 5 and (b) a BNN with 8 features, 2 layers and a hidden dimensionality of 64.

Hamiltonian Monte Carlo algorithm. We plot the performance of the different methods in Figure 4b. Our model can approximate the PPD on the prior much closer than both of these methods, and its inference is more than $200\times$ faster than MLE-II and $1\,000\times$ to $8\,000\times$ faster than NUTS. Further details can be found in Appendix F.

## 5.3 APPROXIMATING BNNS

Bayesian Neural Networks (BNNs) provide a strong ability to model uncertainty due to their Bayesian nature, compared to traditional neural networks. Recently, posterior approximation methods, such as stochastic variational inference (SVI) and stochastic gradient MCMC methods (such as NUTS) allow for much faster convergence, which has sparked a lot of interest in the area (Izmailov et al., 2021; Fortuin et al., 2021). However, computational feasibility still remains a key issue of current BNNs.

In Figure 5, we show that our PFNs can outperform today's default methods for BNN inference using a fraction of their compute budget (PFNs are $1\,000\times$ faster than Bayes-by-Backprop SVI and $10\,000\times$ faster than NUTS to achieve the same performance). The figure shows the Prior-Data NLL of the BNN prior that is used during inference by all solvers. To generate prior-data for our PFNs, we sampled random weights for a $\Phi$ and used normally-distributed i.i.d. feature vectors $X_i$ to obtain $y_i := \Phi(X_i)$ with $\mathcal{D} := (X_{1:n}, y_{1:n})$. For the evaluation we sampled 100 datasets, with $m = 200$ test data points and $n = 100$ revealed data points per dataset. A detailed description of baselines as well as further experiments are included in Appendix F.

## 6 APPLICATION TO TABULAR DATASETS

We evaluate Transformer-based classifiers with priors over Gaussian Processes and Bayesian Neural Networks on a range of small real-world tabular datasets. Surprisingly, the simple priors we describe next outperform a large set of baselines, including XGB (Chen & Guestrin, 2016) and Catboost (Prokhorenkova et al., 2018), while greatly surpassing all baselines in uncertainty calibration.

**A GP Prior for Tabular Classification** We build a classification prior based on GPs with a hyperprior, as described in Section 5.2. Before, we used this prior for regression. To make it a classification prior, we find the median over all targets in the dataset $y$ and set the class to one if greater than the median and zero otherwise.

**A BNN Prior over Architectures** We generalize our BNN setup to learn a prior over model architectures. Fitting a neural network requires finding a suitable architecture, e.g., embedding size, number of layers and activation function. Commonly, resource-intensive searches need to be employed to find suitable architectures (Liu et al., 2018; Zoph et al., 2018). The result of this search is not a posterior distribution over architecture choices and their respective parameters, though, but instead only a point-estimate of the architecture choice. This is true for Bayesian optimisation approaches as

**(a)** **(b)**

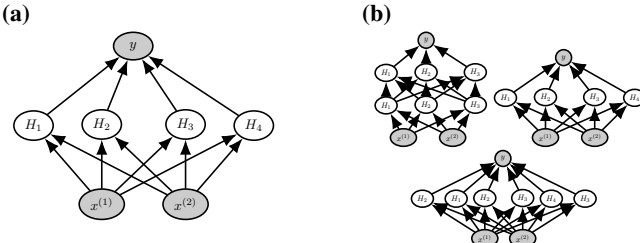

Figure 6: (a) A BNN. The inputs $x$ are mapped to the output $y$. (b) A prior over BNN architectures. The network considers a distribution over different model architectures at once.

well (Kandasamy et al., 2019; Zhou et al., 2019; Ru et al., 2021). Ensembling methods over multiple architectures can yield a rough approximation to a distribution over architectures, but this scales linearly in cost with the number of architectures considered (Zaidi et al., 2020). Using dropout can be regarded as ensemble learning but is limited to varying connectivity (Hara et al., 2017). PFNs allow us to be fully Bayesian, not only about the weights of the BNN, but its architecture, too. By defining a prior probability over the space of model architectures, we remove the need for finding an optimal neural architecture and instead jointly integrate over the space of architectures and their respective weights in a single Bayesian inference. This yields a distribution over architectures in the posterior. The degrees of freedom of model architectures we considered can be found in Appendix G. Our prior-data sampling algorithm for learning to predict based on a prior over both weights and architectures iterates the following steps:

(1) Sample a model architecture $A \sim p(A)$
(2) Sample model weights for the given architecture: $W_{i,j} \sim p_w(\cdot)$
(3) Sample i.i.d. features $x_{i,f} \sim \mathcal{N}(0,1)$ for each data point $i$ and each feature $f$
(4) Propagate each $x_i$ forward through $A_W$ yielding $\{(x_i, A_W(x_i))\}_{i=1}^{N}$

We binarized targets like for the GP Prior above. See Appendix G for more details.

**Setup of Datasets** We used a large collection of tabular datasets from the open-source OpenML AutoML Benchmark (Gijsbers et al., 2019); we first removed datasets with more than one hundred features or missing values, ending up with 20 datasets that represent a diverse set of classification problems with numerical and categorical features. We then simplified each dataset to a balanced binary classification problem, and, for each dataset sampled 20 subsets, each including 100 samples. Within each subset we provide labels for the first 30 samples and evaluate on the remaining samples. All baselines and PFNs use the same datasets and subsets. We also define a set of six unrelated validation datasets used for optimizing the prior distribution over architectures of the PFNs. This is similar to setting the range of hyperparameters in a cross-validation grid search and can be reused for all similar problems. See Appendix G for more details.

**Setup of Models** Our baselines comprise two Gradient Boosted Tree algorithms, XGBoost (Chen & Guestrin, 2016) and Catboost (Prokhorenkova et al., 2018), as well as logistic regression, K-Nearest Neighbors, Gaussian Processes (RBF Kernel) and Bayesian Neural Networks (Bayes-by-Backprop SVI with Pyro). Catboost additionally received information on which features are categorical. We used two PFNs, one PFN for each prior described above. We were able to apply the same model across different feature dimensions. We used grid search with 5-fold cross-validation to optimize our baselines' hyperparameters for each dataset, using the hyperparameter spaces described in Table 6 in the appendix.

**Experimental Results** Table 1 compares the area under the receiver operating curve (ROC AUC) of PFNs based on BNN and GP priors against the various baselines. While the PFN-GP demonstrates on-par performance to the best of the evaluated baselines, the PFN-BNN performs strongest in this comparison. The Wilcoxon signed-rank test (Demšar, 2006), a standard metric for comparing classifiers across multiple datasets is evaluated on each of the 20 subsets of the 21 evaluation datasets separately and shows a significant performance improvement. While all baselines make use of cross-validation to find suitable hyperparameters, the PFNs naturally consider a distribution of hyperparameters and integrate cross-validation and training within the model's Bayesian inference step. As shown in Table 1, this makes it very fast compared to the baselines: since all it needs to do on a new dataset is a single forward propagation step, when run on a GPU (Nvidia Tesla V100), it requires as little as 13 seconds for all 20 datasets combined. This is more than $5000\times$ faster than the 20 hours XGBoost requires, and even faster than K-Nearest Neighbors.

| Metric | PFN-BNN | PFN-GP | Log. Reg. | GP | KNN | BNN | Catboost | XGB |
|---|---|---|---|---|---|---|---|---|
| Mean rank ROC AUC | **2.786** | 3.833 | 4.690 | 5.286 | 6.214 | 5.000 | 4.833 | 3.357 |
| Loss/Tie/Win vs PFN-BNN | – | 14/1/6 | 16/1/4 | 17/0/4 | 17/1/3 | 17/1/3 | 13/1/7 | 12/2/7 |
| Wilcoxon p. vs PFN-BNN | – | 3.6e-13 | 3.6e-13 | 3.9e-13 | 3.6e-13 | 3.6e-13 | 1.9e-12 | 1.6e-06 |
| Expected Calibration Error | **0.025** | 0.067 | 0.157 | 0.095 | 0.093 | 0.089 | 0.157 | 0.066 |
| Benchmark Time | GPU: **0:00:13** CPU: 0:04:23 | | 0:09:56 | 0:24:30 | 0:00:34 | 12:04:41 | 2:05:20 | 20:59:46 |

Table 1: Aggregated performance on subsets of datasets with 30 training samples. Our novel PFN-BNN performs strongest overall. See Table 7 in the appendix for per-dataset results.

We can also experimentally confirm that our PFNs make well-adjusted uncertainty predictions. We find that on our data they have lower Expected Calibration Error (ECE, using 100 bins (Naeini et al., 2015)) even than standard BNNs and ensemble methods such as XGBoost, the two baseline methods that best capture uncertainty. We also provide confidence histograms in Figure 10 of the appendix, visually demonstrating the PFN's strong calibration.

# 7 APPLICATION TO FEW-SHOT LEARNING

As shown in the previous section, PFNs can be trained with priors that are not (easily) possible to use in the traditional Bayesian setups. In this section, we show another example of this, building a prior for hand-written digits and letters and demonstrating strong performance on Omniglot (Lake et al., 2015) in combination with fine-tuning of the PFN on Omniglot.

As an approximation to hand-written symbols, we create a prior that generates random straight lines. In each sample from this prior, we consider five different random line assortments as class prototypes. For each instance of a class, we apply noise to the positions of the lines. It required just 55 lines of readable code to create this prior (see Figure 11 in the Appendix). In Figure 12 of the Appendix we show a sample dataset from this prior.

For our evaluation on Omniglot, we follow the setup of Rothfuss et al. (2021). We perform 5-shot 5-way classification, i.e. we consider five classes and five examples of each class. We create one task from the first five classes of each of the 30 train and 20 evaluation alphabets in Omniglot. We train our PFN on the synthetic prior and then fine-tune it on the 30 training tasks created. This practice is unlike the previous sections. Our premise is to show the flexibility of our model: it can even be fine-tuned on similar datasets, if these are available. We do not use task augmentation, like Rothfuss et al. (2021), but apply translations to each image, as we only use simple linear layers to encode the images in the Transformer. In Table 2 one can see that for this very hard version of the Omniglot dataset our simple method yields results on par with the state of the art. For more information on the setup, please see Appendix H.

| Method | Accuracy | Method | Accuracy |
|---|---|---|---|
| Vanilla BNN (Liu & Wang, 2016) | $0.795 \pm 0.006$ | MLAP (Amit & Meir, 2018) | $0.700 \pm 0.0135$ |
| MAML (Finn et al., 2017b) | $0.693 \pm 0.013$ | BMAML (Yoon et al., 2018) | $0.764 \pm 0.025$ |
| PACOH-NN (Rothfuss et al., 2021) | $\mathbf{0.885 \pm 0.090}$ | PFN (this work) | $\mathbf{0.865 \pm 0.019}$ |

Table 2: Comparison of meta-learning algorithms in terms of test accuracy on the Omniglot dataset. Results within the confidence interval of the best performance are bold-faced.

# 8 CONCLUSION & FUTURE WORK

We presented Prior-Fitted Networks (PFNs), a novel way to efficiently approximate the posterior predictive distribution using deep neural networks and showed its capability on a set of diverse tasks and priors. We expect the following future work to be especially fruitful: (i) Work on finding novel priors that are now feasible to approximate using PFNs. (ii) Work on architectures that are well-fit for this task, as we simply used a slight adaption of current Transformer models. (iii) Work on scaling PFNs to even larger real-world problems. (iv) Work on using our model for the amortized simulation-based inference setting.

## 9 ACKNOWLEDGEMENTS

This research was funded by the Deutsche Forschungsgemeinschaft (DFG, German Research Foundation) under grant number 417962828. Robert Bosch GmbH is acknowledged for financial support. We also want to thank the maintainers of PyTorch (Paszke et al., 2019).

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

## A    PROOF OF COROLLARY 1.1

*Proof.* This can easily be seen by considering the result of Insight 1 for the last transformation in line 9.

$$\mathbb{E}_{x,D}[KL(p(\cdot|x,D), q_\theta(\cdot|x,D))] \tag{5}$$

$$= -\mathbb{E}_{x,D}\left[\int_y p(y|x,D)\log\frac{q_\theta(y|x,D)}{p(y|x,D)}\right] \tag{6}$$

$$= -\mathbb{E}_{x,D}\left[\int_y p(y|x,D)\log q_\theta(y|x,D)\right] + \mathbb{E}_{x,D}\left[\int_y p(y|x,D)\log p(y|x,D)\right] \tag{7}$$

$$= \mathbb{E}_{x,D}[\mathrm{H}(p(\cdot|x,D), q_\theta(\cdot|x,D))] - \mathbb{E}_{x,D}[\mathrm{H}(p(\cdot|x,D))]] \tag{8}$$

$$= \ell_\theta + C, \tag{9}$$

where $C = -\mathbb{E}_{x,D}[\mathrm{H}(\cdot|x,D)]$ is a constant that does not depend on $\theta$. $\square$

## B    PROOF OF COROLLARY 1.2

*Proof.* We assume a $\theta$ with $q_\theta = p$ exists and we know that the cross-entropy is minimized for equal distributions; thus the optimum $\theta^*$ fulfills $q_{\theta^*}(\cdot|x,D) = p(\cdot|x,D)$ for all $x,D$ where the density of the prior fulfills $p(x,D) > 0$, which is exactly the subset for which the conditional is defined. $\square$

## C    PROOF THEOREM 1

*Proof.* Given an arbitrary distribution $p$, an arbitrary bound $\epsilon$. For arbitrary minimum (maximum) bucket borders $a$ ($b$), we consider the function $b_\mathbf{B}$ that maps each position $y$ to its bucket $b \in \mathbf{B}$, where $\mathbf{B}$ is an arbitrary set of buckets. We define the upper bound of $p$ for each bucket

$$u(y) = \sup_{y'\in b_s(y)} p(y'). \tag{10}$$

Similarly, we define the lower bound

$$l(y) = \inf_{y'\in b_s(y)} p(y'). \tag{11}$$

For each of these we consider their log form, too. That is $\hat{u}(y) = \log u(y) = \sup_{y'\in b_s(y)} \log p(y')$ and $\hat{l}(y) = \log l(y) = \inf_{y'\in b_s(y)} \log p(y')$. We can move the log inside the sup (inf), since log is steadily increasing.

Integrals over $\hat{u}$ and $\hat{l}$ thus handily represent Darboux's upper and lower sums of $\log p$, if $\log p$ is Riemann integrable and finite. We assumed that $p$ is Riemann integrable, thus $\log p$ is also integrable, since the composition of continuous functions, which $\log x$ is for $x > 0$, with an almost everywhere continuous function is almost everywhere continuous and any finite value $x$ has a finite value $\log x$, if $x > 0$. The requirement $x > 0$ is given due to the assumed full support of $p$. Thus, we can use $\forall\epsilon'\exists\mathbf{B}. \int_a^b \hat{u}(y) - \hat{l}(y)dy < \epsilon'$. We define $t = \sup_{y\in[a,b]} p(y)$, $T = \sup_{y\in\mathbb{R}} p(y)$ and the total probability mass of the interval $P = p(y \in [a,b])$. We use $\epsilon' = \log 1/P$, which fulfills $\epsilon' > 0$ since $p$ has full support and thus for any borders $a$ and $b$ we have $P < 1$, get $\mathbf{B}$ s.t.

$$\int_a^b \hat{u}(y) - \hat{l}(y)dy < \log\frac{1}{P} \tag{12}$$

$$\Rightarrow \int_a^b t(\hat{u}(y) - \hat{l}(y))dy < t\log 1/P \tag{13}$$

$$\Rightarrow \int_a^b t\log\frac{u(y)}{l(y)}dy < t\log 1/P \tag{14}$$

$$\Rightarrow \int_a^b u(y)\log\frac{u(y)}{l(y)}dy < t\log 1/P \tag{15}$$

Let $c$ be a constant s.t. $cu(y)$ yields a valid probability distribution with $\int cu(y)dy = \int_a^b cu(y)dy = 1$.

$$\Rightarrow \int_a^b cu(y) \log \frac{u(y)}{l(y)} dy < ct \log 1/P \tag{16}$$

$$\Rightarrow \int_a^b cu(y) \log \frac{cu(y)}{l(y)} dy < ct \log 1/P + \int_a^b cu(y) \log c \tag{17}$$

$$\Rightarrow \int_a^b cu(y) \log \frac{cu(y)}{l(y)} dy < ct \log 1/P + \log c \tag{18}$$

We use $c < 1/P$, as $1/P$ would be the factor $c'$ to make $\int_a^b c'p(y)dy = 1$ and $c$ is this factor for $u(y)$, which has $u(y) \geq p(y)$ for any $y \in [a, b]$.

$$\Rightarrow \int_a^b cu(y) \log \frac{cu(y)}{l(y)} dy < ct \log 1/P + \log 1/P \tag{19}$$

$$\Rightarrow \int_a^b cu(y) \log \frac{cu(y)}{l(y)} dy < (ct + 1) \log 1/P \tag{20}$$

$$\Rightarrow \int_a^b cu(y) \log \frac{cu(y)}{l(y)} dy < -t\frac{\log P}{P} - \log P \tag{21}$$

$$\Rightarrow \int_a^b cu(y) \log \frac{cu(y)}{l(y)} dy < -T\frac{\log P}{P} - \log P \tag{22}$$

$$\Rightarrow \int_a^b cu(y) \log \frac{cu(y)}{p(y)} dy < -T\frac{\log P}{P} - \log P \tag{23}$$

$$\Rightarrow KL(cu(y), p(y)) < -T\frac{\log P}{P} - \log P \tag{24}$$

$$\Rightarrow KL(cu(y), p(y)) < -(T + P)\frac{\log P}{P} \tag{25}$$

$$\Rightarrow KL(cu(y), p(y)) < -(T + 1)\frac{\log P}{P}. \tag{26}$$

Here, $cu(y)$ is a valid bar distributions. Now we want to find $a$ and $b$ such that $-(T+1)\frac{\log P}{P} < \epsilon$, which is equivalent to

$$\frac{1}{P} \log 1/P < \frac{\epsilon}{T + 1} \tag{27}$$

Here, we can substitute the Lambert W function to arrive at the final result, that we should choose $a$ and $b$ such that $p([a, b]) \geq e^{-W_0(\frac{\epsilon}{T+1})}$, which is always possible as $W_0$ maps to $(0, \infty)$ for positive inputs, which we have with $\frac{\epsilon}{T+1}$. And we have that $e^{-x} < 1$ and $e^{-x} > 0$ for any $x \in (0, \infty)$.

Thus, we can conclude the proof. We could upper bound the *KL*-Divergence for any $a$ and $b$. Then, based on that bound we found a setting for values $a$ and $b$ such that this upper bounds falls below $\epsilon$.

$\square$

## D  GENERAL TRANSFORMER TRAINING DETAILS

We follow the standard Transformer encoder setup. In the whole discussion one should note the crucial difference to a standard MLE training: Our optimization problem does not allow for overfitting. An improvement in training loss always means an improvement in the closeness to the exact posterior. More training just improves performance and calibration. The only exception of this is the fine-tuning done for the few-shot learning experiment in Section 7. We use Adam (Kingma & Ba, 2015) and cosine decay (Loshchilov & Hutter, 2016) with warmup. For all experiments we used a embedding size of 512, only for few-shot classification we used 1024. The only hyper-parameters that we did fine-tune for the Transformer training were the batch size and learning rate. Although, it usually was enough to choose a large enough batch size while the learning rate was more important. Learning

rates generally where chosen based on training performance, if not specified differently. We can do this as we generate new data during training and thus do not need to tune on held-out data. We saw that search over the set $\{.00001, .0001, .0003, .001\}$ for a learning rate generally yielded good enough results. Usually performance of the models got better with more training, thus the number of samples used for training generally depends on our compute budget. For setups models considered we could improve performance with more training. We stopped when we were satisfied with the results. We did not train longer than a few days on a single standard GPU for any PFN.

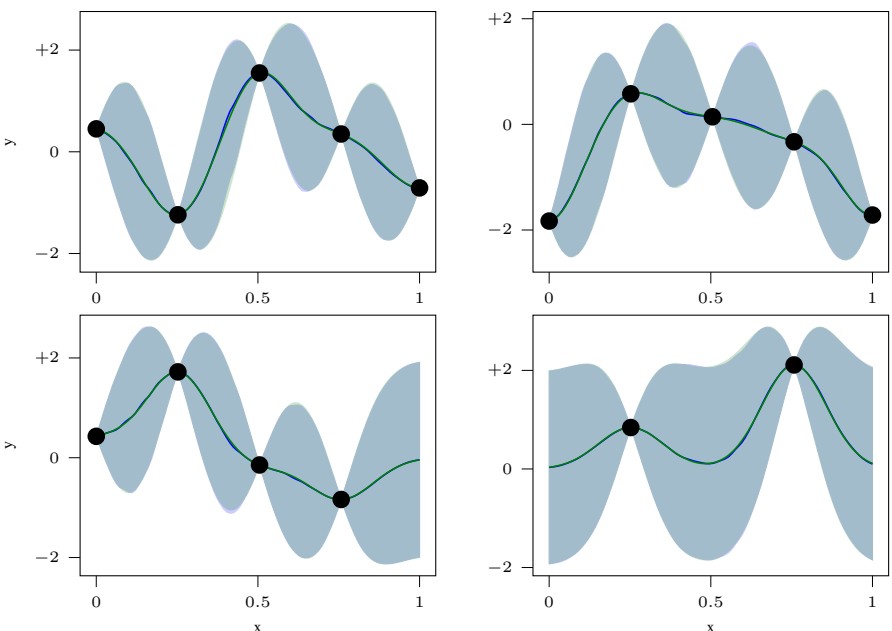

Figure 7: We show the mean and 95% intervals of the original GP posterior in green and the corresponding PFN approximations in blue. We used a length scale of $0.1$.

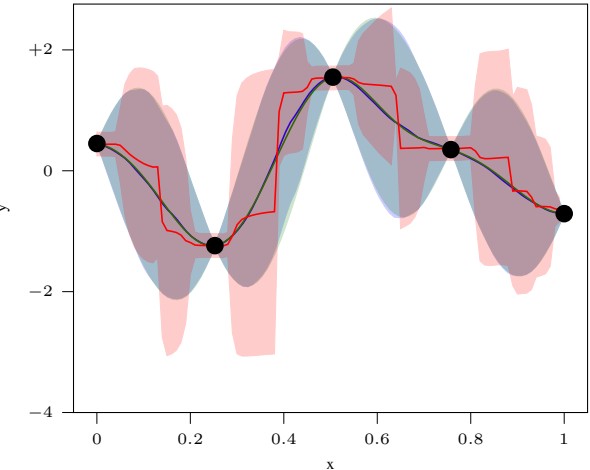

Figure 8: Like in plot 7, we show the mean and 95% confidence intervals for the exact GP prior in green and for our PFN in blue. (They are almost indistinguishable, with very small differences visible, e.g., at $x = 0.4$.) To further highlight what a big step forward our method is, we additionally show the respective values for the relatively new Attentive Neural Processes (ANP) (Kim et al., 2018). We trained both methods in very different environments, but ANP was clearly given the bigger budget.

# E  TRANSFORMER ADAPTIONS

## E.1  AN EFFICIENT ARCHITECTURE

The first step of the model is to encode $x$ and $y$; in all our experiments, we use simple linear layers for this, but one could also use other application-specific encoding layers. Our setup is equivariant to the ordering of the input dataset's examples. To achieve this, we remove positional encodings and feed $(x, y)$ pairs together to the Transformer as a sum of their encodings; see Appendix E.3 for an ablation of the impact of the equivariance to the ordering. In practice, we feed query points together for efficiency reasons; these are the only inputs for which position matters, and we thus use the output at their positions as prediction for their corresponding PPD over $y$. Since different query points should not depend on each other, we use an attention mask that allows attention from every input example to every other input example and allows attention from every query points to every input example. An example of the model and its attention can be seen in Figure 2a.

During training, we fix the total number of inputs $N = n + m$, where $n$ is the number of inputs and $m$ is the number of queries, and sample $n$ from a distribution, in order to allow the Transformer to learn to work with different dataset sizes. Since for smaller $n$ we have more query points to train on, we over-sample larger $n$, s.t. the number of query points seen during training for each $n$ is approximately equal. That is, we sample each $n \in \{0, \ldots, N - 1\}$ with a weight $1/m = 1/(N - n)$. As prediction head we use a softmax for multi-class classification tasks, a sigmoid for binary classification tasks and the Riemann distribution for regression.

## E.2  RIEMANN DISTRIBUTION DEFINITION

Given a set of of buckets $\mathbf{B}$, where each bucket $b \in \mathbf{B}$ describes an interval such that

$$\bigcup_{b \in \mathbf{B}} b = [a, b] \tag{28}$$

and for any $b, b' \in \mathbf{B}$, we have $b \cap b' = \varnothing$. Additionally, we define the upper-bound (lower-bound) of the lowest (highest) bucket to be $l$, and a function $B(\cdot)$ that maps a $y \in \mathbb{R}$ to the unique bucket $b$ with $y \in b$. We assume to have a function $w$ mapping to the width of each bucket as $w(b) = \max_{y \in b} y - \min_{y \in b} y$. Lastly, we assume the model gives us a probability $p_b$ for each bucket $b$.

In the finite case the Riemann distribution is defined as

$$p(y) = p_{b(y)}/w(b(y)). \tag{29}$$

We normalize with the width of the bucket since $p_b$ describes the probability $p(y' \in b)$, but we are interested in the probability $p(y = y')$.

For the case of infinite support, we have

$$f(y_q) = \begin{cases} 2 \cdot f_{\mathcal{N}}(y_q - l), & \text{if } y_q \geq l \\ 2 \cdot f_{\mathcal{N}}(f - y_q), & \text{if } y_q \geq f \\ p_{b(y_q)}/b(y_q), & \text{otherwise.} \end{cases} \tag{30}$$

Here, the Half-Normal distributions are scaled such that half the probability weight is inside the bounds of the first (last) bucket.

## E.3  ABLATION STUDIES

Above, we introduce a novel way of doing regression with neural networks that departs from the tradition of using normal distributions. Additionally, we remove the default positional embeddings from the Transformer. We show the impact of both approaches on the Transformer in Figure 9. Notice that, while the effect of the Riemann Distribution is very pronounced, the effect of making our architecture permutation invariant only slowly increases with sequence length. For each setup we searched over a set of the same four learning rates and evaluated the best performing on another sample from the distribution.

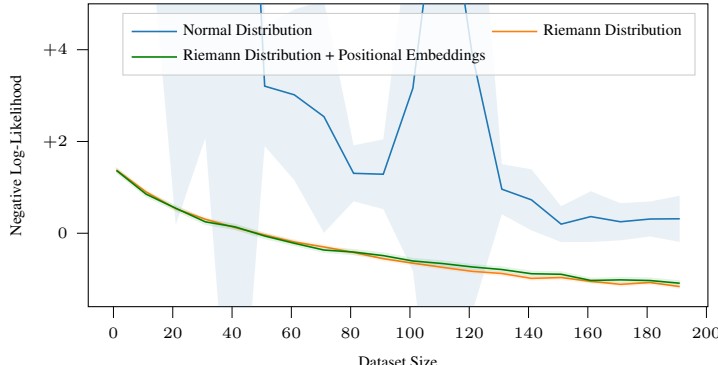

Figure 9: In this figure we show the impact of different architectural decisions on the performance of our method in fitting a Gaussian Process. We use the same Gaussian Process as in Section 5.1.

## F    DETAILS FOR SECTION 5

The error intervals that can be seen in the Prior-Data NLL plots are 95% percent confidence intervals over different dataset samples from the prior.

**Details for Section 5.1**    Here, we generally followed default settings for GPs. That is, we used the RBF-Kernel with zero-mean, a length scale of 0.6, 1e-4 additional noise on the diagonal of the covariance matrix. The output scale was set to 1. Only for the plots in the Figure 7 and Figure 8 the length scale was decreased to 0.1. The experiments with more examples in Figure 4a did use 5 input dimension, while the qualitative visualizations only used one. For all methods we sample 1000 datasets for evaluation, each of which has a single validation example, for each dataset size.

**Details for Section 5.2**    In this section, we followed the default settings that can be found in BoTorch's *SingleTaskGP* (Balandat et al., 2020). We only changed the distribution over the diagonal noise, as the default is very high, such that it is hard to see any improvements with any of the methods in NLL with an increasing number of training samples. The GP is setup as a Matérn Kernel with $\nu = 2.5$ and we have the following hyper-priors: the noise was sampled from $\Gamma(\alpha = 0.0001, \beta = 1.0)$, the output scale from $\Gamma(\alpha = 2.0, \beta = 0.15)$ and the length scale from $\Gamma(\alpha = 3.0, \beta = 6.0)$.

We used the Pyro (Bingham et al., 2018) implementation of NUTS. Here, we used every sampled hyper-parameter setting as a sample GP. We averaged the output distribution over all sampled GPs to yield an approximation to the PPD. We scaled both the warmup and the rollout steps together: with "8 (warmup) steps" we mean 8 warmup steps and then 8 sampling steps.

Only the MCMC experiments were run on CPU, as this was faster than using the GPU. For all methods we sample 1000 datasets for evaluation, each of which has a single validation example, for each dataset size.

**Details for Section 5.3**    For SVI (Bayes-by-Backprop) we followed the setup of (Blundell et al., 2015) and used Adam (Kingma & Ba, 2015) as optimizer. We leveraged Pyro (Bingham et al., 2018) for optimization. We scaled both the training steps and number of samples averaged together to increase approximation quality.

For MCMC we used the Pyro (Bingham et al., 2018) implementation of NUTS and averaged the output distribution over all sampled BNNs to yield an approximation to the PPD. We scaled both the warmup and the rollout steps together to increase approximation performance. We used one MCMC chain. PFN and SVI are evaluated on a GPU (Nvidia Tesla V100) while MCMC is evaluated on a CPU since it was faster on CPU than on GPU.

**(a) Bayesian Neural Network**   **(b) PFN-BNN**   **(c) XGBoost**

Figure 10: Comparison of uncertainty calibration curves over all predictions on the evaluated datasets. The dotted line indicates perfect calibration.

## G  DETAILS FOR SECTION 6

**Accomodating a variable number of features** Since the evaluated datasets used a variable number of features, the dimensionality of the input is variable. This can not be accommodated by the fixed size of the linear layer used in our PFNs. Thus, we fixed a maximum number of features (60) and extended the datasets to this maximum number by appending zeros. We did this during prior training, sampling the number of features uniformly at random alongside the model architecture and during inference. We rescale the features by a factor of $\frac{\text{max features}}{\text{features used}}$ during prior training and bayesian inference to keep the mean input scale constant with a varying feature number.

**Degrees of freedom for model architectures** The evaluated BNN models consider a wide range of model architectures, that is those architectures vary in terms of: number of layers, number of hidden units per layer, layer sparsity, standard deviation of Gaussian noise applied to each unit and activation function applied to each unit. This list can be easily extended by adapting the sampling procedure.

**Tuning hyperparameter ranges for hyper-priors and cross-validation** While the Transformer-based methods need no parameter tuning on each dataset individually, they still require setting the prior hyperparameters. This is similar to setting the range of hyperparameters in a cross-validation grid search and can be reused for all similar problems, e.g., all small-scale tabular classification. We therefore run a hyperparameter search on a second set of validation datasets, listed in Table 4. The hyperparameters considered in this grid search can be found in 5 In this search, we find good settings for all hyper-parameters of our priors for the average over all validation datasets. This search results in one final hyper-parameter setting and only one single Transformer model that is reused for all datasets. We use the same setup to find cross-validation ranges for our baselines.

**Standardization of datasets** Each subset is standardized to have zero mean and unit variance where the statistics for the standardization are only calculated on the samples considered by the model, thus without any dependence on the test set.

**Runtimes of prior fitting step** Fitting our transformer with GP prior was done over 100 epochs, taking 2:44h, while fitting the BNN prior took 3:14h for the same number of epochs. Crucially, this model is reused for all datasets.

## H  DETAILS FOR SECTION 7

We searched for the hyper-parameters specified as inputs in Figure 11 with a grid search, where we measure performance on the training tasks as validation set. We then trained our model as a PFN on samples from this prior. After this stage we fine-tuned the model on the training tasks of Omniglot. Here, we applied translations and used a default batch size of 100. We used a learning rate of 1e-5 to yield the performance shown in the plot after sampling 500,000 samples from the training tasks. Unlike above, we sample over different samples of our fine-tuned model to yield the confidence interval, to follow the standard reporting on the Omniglot benchmark.

```python
def pencil_stroke_prior(num_classes=2, size=28, min_max_strokes=(1,3),
        min_max_len=(5/28,20/28), min_max_start=(2/28,25/28),
        min_max_width=(1/28,4/28), max_offset=4/28, max_target_offset=2/28):
    classes = []
    for i in range(num_classes):
        num_strokes = random.randint(*min_max_strokes)
        len_strokes = [random.randint(int(size * min_max_len[0]),
                        int(size * min_max_len[1]))
                        for i in range(num_strokes)]
        stroke_start_points = [
            (random.randint(int(size * min_max_start[0]), int(size * min_max_start[1])),
            random.randint(int(size * min_max_start[0]), int(size * min_max_start[1])))
            for i in range(num_strokes)]
        stroke_directions = []
        # i = Image.fromarray(np.zeros((28,28), dtype=np.uint8))
        # draw = ImageDraw.Draw(i)
        for i in range(num_strokes):
            sp, length = stroke_start_points[i], len_strokes[i]
            counter = 0
            while True:
                if counter % 3 == 0:
                    length = random.randint(int(size * min_max_len[0]),
                        int(size * min_max_len[1]))
                    sp = (
                    random.randint(int(size * min_max_start[0]),
                        int(size * min_max_start[1])),
                    random.randint(int(size * min_max_start[0]),
                        int(size * min_max_start[1])))
                    stroke_start_points[i], len_strokes[i] = sp, length
                radians = random.random() * 2 * math.pi
                x_vel = math.cos(radians) * length
                y_vel = math.sin(radians) * length
                new_p = (sp[0] + x_vel, sp[1] + y_vel)
                if not any(n > size - 1 or n < 0 for n in new_p):
                    break
                counter += 1
            stroke_directions.append(radians)
        classes.append((len_strokes, stroke_start_points, stroke_directions))
    generator_functions = []
    for c in classes:
        def g(c=c):
            len_strokes, stroke_start_points, stroke_directions = c
            i = Image.fromarray(np.zeros((size, size), dtype=np.uint8))
            draw = ImageDraw.Draw(i)
            width = random.randint(int(size * min_max_width[0]),
                        int(size * min_max_width[1]))
            offset = (random.randint(int(-size * max_offset), int(size * max_offset)),
                    random.randint(int(- size * max_offset), int(size * max_offset)))
            for sp, length, radians in \textbackslash
                    zip(stroke_start_points, len_strokes, stroke_directions):
                sp = (sp[0] + offset[0], sp[1] + offset[1])
                x_vel = math.cos(radians) * length
                x_vel += random.randint(int(-size * max_target_offset),
                                    int(size * max_target_offset))
                y_vel = math.sin(radians) * length
                y_vel += random.randint(int(-size * max_target_offset),
                                    int(size * max_target_offset))
                new_p = (sp[0] + x_vel, sp[1] + y_vel)
                stroke_directions.append(radians)
                draw.line([round(x) for x in sp + new_p], fill=128, width=width)
            a_i = np.array(i)
            a_i[a_i == 128] = np.random.randint(200, 255, size=a_i.shape)[a_i == 128]
            return Image.fromarray(a_i).filter(ImageFilter.GaussianBlur(.2))
        generator_functions.append(g)
    return generator_functions
```

Figure 11: A prior definition for a prior over straight strokes in python.

| Name | Number of Features | Number of Symbolic Features |
|---|---|---|
| kr-vs-kp | 37 | 37 |
| credit-g | 21 | 14 |
| vehicle | 19 | 1 |
| wine | 14 | 1 |
| kc1 | 22 | 1 |
| airlines | 8 | 5 |
| bank-marketing | 17 | 10 |
| blood-transfusion-service-center | 5 | 1 |
| phoneme | 6 | 1 |
| covertype | 55 | 45 |
| numerai28.6 | 22 | 1 |
| connect-4 | 43 | 43 |
| car | 7 | 7 |
| Australian | 15 | 9 |
| segment | 20 | 1 |
| jungle_chess_2pcs_raw_endgame_complete | 7 | 1 |
| sylvine | 21 | 1 |
| MiniBooNE | 51 | 1 |
| jannis | 55 | 1 |
| helena | 28 | 1 |

Table 3: Tabular datasets used for evaluation. Taken from the OpenML AutoML Benchmark (filtered for $N_{features} \leq 100$, no missing values.)

| Name | Number of Features | Number of Symbolic Features |
|---|---|---|
| haberman | 4 | 2 |
| ionosphere | 35 | 1 |
| sa-heart | 10 | 2 |
| cleve | 14 | 9 |
| four | 2 | 0 |
| german | 24 | 0 |

Table 4: Tabular datasets used to find good general hyperparameters for hyper-priors as well as hyperparameters used in baseline cross-validation.

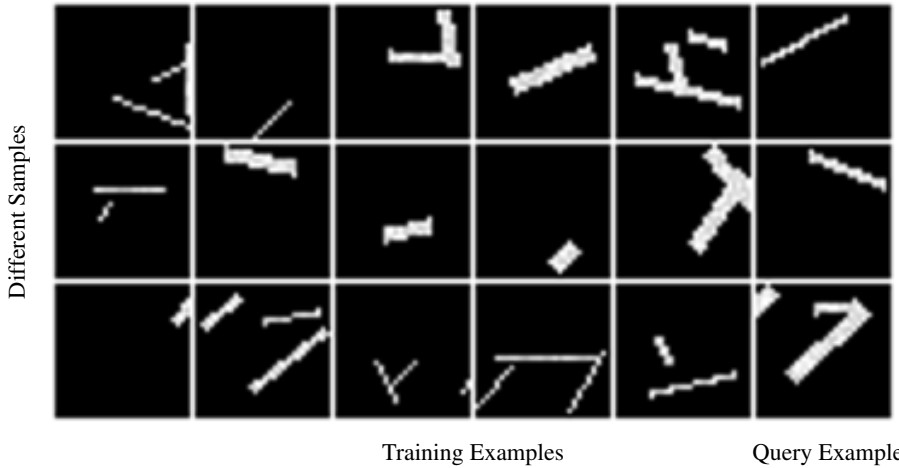

Figure 12: In this figure we show different samples from our prior for Omniglot data.

| index | n | param |
|---|---|---|
| prior_activations | 1 | torch.nn.Tanh |
| | 2 | torch.nn.ReLU |
| prior_dropout_sampler | 1 | lambda: 0.0 |
| | 2 | lambda: 0.5 |
| | 3 | beta_sampler_f(1.1, 1.4) |
| | 4 | beta_sampler_f(1.1, 2.0) |
| | 5 | beta_sampler_f(1.1, 4.0) |
| | 6 | beta_sampler_f(2.0, 1.4) |
| | 7 | beta_sampler_f(2.0, 4.0) |
| prior_emsize_sampler | 1 | scaled_beta_sampler_f(1.1, 1.3, 256, max_features) |
| | 2 | scaled_beta_sampler_f(1.1, 2.0, 256, max_features) |
| | 3 | scaled_beta_sampler_f(1.1, 4.0, 256, max_features) |
| | 4 | scaled_beta_sampler_f(2.0, 4.0, 100, 1+max_features) |
| prior_nlayers_sampler | 1 | lambda: 3 |
| | 2 | scaled_beta_sampler_f(1.1, 1.3, 4, 3) |
| | 3 | scaled_beta_sampler_f(1.1, 2.0, 5, 2) |
| | 4 | scaled_beta_sampler_f(1.1, 4.0, 5, 2) |
| | 5 | scaled_beta_sampler_f(2.0, 4.0, 4, 3) |
| prior_noise_std_gamma_k | 1 | range: [1.5, 5.0] |
| prior_noise_std_gamma_theta | 1 | range: [0.01, 1.0] |
| prior_sigma_gamma_k | 1 | range: [1.5, 5.0] |
| prior_sigma_gamma_theta | 1 | range: [0.01, 1.0] |

Table 5: Hyperparameters considered during grid search tuning of the PFN-BNN on validation datasets. The activation functions refer to the activation function used in the data generating BNN and not the ones used in the PFN transformer. scaled_beta_sampler_f(a,b,max,min) refers to sampling from a beta distribution with parameters a, b (numpy.random.beta), scaling the output range from [0, 1] to [max, min] and then rounding the output to the integer range. beta_sampler_f(a, b) refers to a function that samples from the beta distribution without scaling and rounding.

| Log. Regr. | {'solver': ['saga'], 'penalty': ['l1', 'l2', 'none'], 'tol': [0.01, 0.0001, 1e-10] , 'max_iter': [500], 'fit_intercept': [True, False], 'C': [1e-05, 0.001, 0.01, 0.1, 1.0, 2.0]} |
|---|---|
| KNN | {'n_neighbors (max number of samples)': array([1, 2, 3, 4, 5])} |
| BNN | {'embed': [5, 10, 30, 64], 'lr': [0.001, 0.0001], 'num_training_steps': [400] , 'num_samples_for_prediction': [400]} |
| GP | {'params_y_scale': [0.05, 0.1, 0.5, 1.0, 5.0, 10.0], 'params_length_scale': [0.1, 0.5, 1.0, 2.0]} |
| Catboost | {'learning_rate': [0.1, 0.5, 1.0], 'depth': [2, 4, 7], 'l2_leaf_reg': [0.0, 0.5, 1] , 'iterations': [10, 40, 70], 'loss_function': ['Logloss']} |
| XGB | {'min_child_weight': [0.5, 1.0], 'learning_rate': [0.02, 0.2], 'subsample': [0.5, 0.8] , 'max_depth': [1, 2], 'colsample_bytree': [0.8], 'eval_metric': ['logloss'], 'n_estimators': [100]} |

Table 6: Hyperparameters used in cross-validation for each baseline method.

| | PFN-BNN | PFN-GP | logistic | GP | KNN | BNN | Catboost | XGB |
|---|---|---|---|---|---|---|---|---|
| kr-vs-kp | 0.928 | 0.897 | 0.920 | 0.886 | 0.754 | 0.843 | 0.653 | **1.000** |
| credit-g | 0.671 | **0.681** | 0.621 | 0.562 | 0.593 | 0.607 | 0.602 | 0.637 |
| vehicle | 0.562 | 0.501 | **0.601** | 0.506 | 0.527 | 0.562 | 0.525 | 0.522 |
| wine | 0.959 | **0.975** | 0.944 | 0.785 | 0.900 | 0.919 | 0.908 | 0.930 |
| kc1 | 0.614 | 0.695 | 0.730 | 0.695 | 0.666 | 0.724 | 0.708 | **0.747** |
| airlines | 0.916 | 0.896 | 0.913 | 0.873 | 0.834 | 0.907 | 0.982 | **0.985** |
| bank-market.. | 0.878 | 0.877 | 0.811 | 0.857 | 0.810 | 0.849 | **0.908** | 0.885 |
| blood-transfus.. | **0.588** | 0.576 | 0.558 | 0.577 | **0.588** | 0.555 | 0.565 | 0.531 |
| phoneme | 0.757 | 0.782 | 0.735 | **0.785** | 0.772 | 0.733 | 0.766 | **0.785** |
| covertype | **0.962** | 0.904 | 0.904 | 0.929 | 0.793 | 0.870 | 0.810 | 0.798 |
| numerai28.6 | 0.449 | 0.454 | 0.460 | **0.502** | **0.502** | 0.454 | **0.502** | 0.464 |
| connect-4 | **0.981** | 0.961 | 0.951 | 0.933 | 0.871 | 0.934 | 0.945 | 0.976 |
| car | **0.959** | 0.926 | 0.925 | 0.771 | 0.874 | 0.935 | 0.849 | 0.898 |
| Australian | **0.906** | 0.903 | 0.862 | 0.866 | 0.865 | 0.873 | 0.848 | 0.871 |
| segment | **1.000** | **1.000** | **1.000** | 0.999 | 0.985 | 0.999 | **1.000** | **1.000** |
| jungle_chess.. | 0.774 | 0.780 | 0.693 | **0.852** | 0.747 | 0.708 | 0.841 | 0.708 |
| sylvine | 0.833 | 0.791 | 0.915 | 0.708 | 0.664 | 0.785 | 0.885 | **0.929** |
| MiniBooNE | 0.855 | 0.852 | 0.821 | 0.843 | 0.850 | **0.861** | 0.818 | 0.855 |
| dionis | **0.996** | 0.977 | 0.981 | 0.982 | 0.962 | 0.987 | 0.975 | 0.991 |
| jannis | **0.654** | 0.652 | 0.570 | 0.551 | 0.569 | 0.549 | 0.603 | 0.600 |
| helena | **0.911** | 0.892 | 0.839 | 0.870 | 0.842 | 0.882 | 0.870 | 0.905 |
| Mean rank AUC | **2.786** | 3.833 | 4.690 | 5.286 | 6.214 | 5.000 | 4.833 | 3.357 |
| Win/Tie/Loss AUC vs T-BNN | 0/21/0 | 14/1/6 | 16/1/4 | 17/0/4 | 17/1/3 | 17/1/3 | 13/1/7 | 12/2/7 |
| Wilcoxon p. AUC vs T-BNN | 0.0e+00 | 3.6e-13 | 3.6e-13 | 3.9e-13 | 3.6e-13 | 3.6e-13 | 1.9e-12 | 1.6e-06 |
| Benchmark Time | GPU: 0:00:13 CPU: 0:04:23 | | 0:09:56 | 0:24:30 | 0:00:34 | 12:04:41 | 2:05:20 | 20:59:46 |
| ECE | **0.025** | 0.067 | 0.157 | 0.095 | 0.093 | 0.089 | 0.157 | 0.066 |

Table 7: ROC AUC Comparison: Evaluation is made on subsets of datasets with 30 training samples. 20 data-subsets are evaluated for each dataset. The ECE is a calibration measure. The novel PFN-BNN performs strongest overall.

