# OpenReview forum: "Transformers Can Do Bayesian Inference"
_ICLR.cc/2022/Conference — ICLR 2022 Poster_

### Official Review · Reviewer_NwoK · 2021-11-02

**Correctness:** 3
**Technical Novelty And Significance:** 2
**Empirical Novelty And Significance:** 4
**Recommendation:** 6
**Confidence:** 4

**Main Review:**

I found the results of this paper pretty impressive, since it shows good performance across many different modalities of data and it is able to generalize well to new datasets. However, I find the main claim of the paper, that "Bayesian inference" is happening, unsubstantiated.

It is true that Fig. 3 shows that the posterior GP and the posterior obtained from the PFN are well matched. This proves that the Bayesian posterior is being recovered _for this particular case_. This is a particularly simple case, and similar results could be obtained simply taking the training data (which is 1D and confined to a small space) and plotting all the training trajectories weighted by how close they are to the two training points when traversing that particular x coordinate. The results would be very similar to those in Fig 3. As we get to more complicated cases, we see that the results from PFN and GPs start to diverge. In Fig 4.b, for 5 data points, we see that the MCMC solution and the PFN solution are different. The fact that PFN performs better in terms of the NLL means that it is a better model for the data, but not necessarily that it's closer to the true posterior. In fact, the MCMC solution, if run long enough, should be the golden standard.

Thus, the provided results show that PFNs perform very well in terms of NLL which means that they are well calibrated, but saying that they are doing Bayesian inference could be a stretch, since outside of the GP with fixed-hyperparameters we have no verification for that (in fact, it seems that either MCMC was not run to covergence, or PFN does not match Bayesian inference).

Minor:

Theorem 1 and the following Corollaries are trivial and that space could probably given a better use. The so-called "Prior data NLL" is just a standard leave-one-out NLL. The connection between maximum likelihood and minimum KL are also trivial and well-known.

Comparisons are missing the training time of the PFN. Although I understand that that time can be amortized, it would be good to have a sense for it in each of the experiments.

How does this process scale with the size of the dataset? Can we use it for GPs with 100.000 training samples for instance?

"we sample the N inputs xi uniformly at random from the unit-cube"-> isn't x_i one dimensional? If so, it's not a unit "cube". What's the value of N?


**Summary Of The Paper:**

In this paper the authors show a simple architecture (simple on top of transformer pieces) that can learn to perform supervised learning on new datasets given collections of existing datasets. The model is trained on a collection of supervised learning datasets, and, at test time a novel training dataset and the corresponding test inputs are given and a single forward pass through the network produces the predictive distributions for those test inputs. Several different types of data are tested with close to state-of-the-art results provided, but in much less time.

**Summary Of The Review:**

Interesting paper and strong results. Probably mainly useful for fast inference in small datasets. The claim of doing Bayesian inference is not fully substantiated beyond 1D GPs.

---

> ### Author Response · Authors · 2021-11-09
> **Reply to Review by Reviewer NwoK**
>
> Dear Reviewer NwoK,
>
> Thank you for reviewing our paper and for your valuable feedback, which we would like to answer in detail now.
>
> **Bayesian Inference**
>
> Regarding the claim of doing Bayesian Inference. We show in Theorem 1, which you rightly pointed out to be quite simple, that our objective actually does yield a posterior approximation in terms of KL-divergence. That is, our method directly allows us to use the quality of the fit to the posterior predictive distribution as our training loss; and by training on more and more artificial datasets from our prior (and making our transformers bigger if needed) we can make this training loss extremely small. Just like variational inference approximates Bayesian inference by minimizing the KL-Divergence to the posterior, we approximate Bayesian inference by minimizing the KL-Divergence with the PPD. Crucially, we also show in our experiments that PPD approximation using PFNs converges much faster than MCMC. We do understand your criticism that MCMC should be the gold standard, but this will require yet much longer chains than we have run so far. We will make sure to include experiments running MCMC much longer in Figure 4b. We are certain that it will improve and converge to similar predictions as PFNs, but this will require days as compared to seconds for the PFNs.
>
> **The Prior-Data NLL**
>
> We would also like to make sure that there are no misunderstanding about the NLL loss:
> In Section 3 we show that the “Prior-Data NLL” equals the KL-divergence with the true PPD up to a constant. So, a lower NLL means we actually approximate the true PPD more closely in terms of KL-divergence. This is exactly what we want to show here.
> Thus, the local dissimilarity between MCMC and PFNs in Figure 4b does not show that PFNs find the wrong solution, as they have the lower NLL, thus approximate the PPD more exactly. It rather shows that MCMC only slowly improves with higher budgets (more steps). MCMC should outperform the exact PFN at some point as you correctly say, but this might require larger budgets than can be afforded in a particular application (10000x higher before reaching the quality of PFNs, as we showed in Figure 5).
>
> **Training time of PFNs**
>
> This is a good point, we will also explicitly state the training time of the PFNs for each experiment. We did not actually consider this much due to the amortization you mention (fitting the PFN is a one-time cost that can be done before seeing any data, so it can be considered part of the algorithm development time), but the runtime is actually not bad.
> Generally the training of the Transformer does not take more than a few hours on a single GPU. Training only required over a day on a single GPU, as we write in the appendix, for the very long sequences of Figure 4a and the fine-tuning for Section 7.
>
> **Scaling**
>
> This is also a very good point, which we briefly discuss in the conclusion of our work. Our current setup can scale up to 4000 examples without any tweaks. Using priors adds performance especially on small datasets so the advantages and limitations converge conveniently here. That is, the usage of a prior benefits tasks with few training samples, but with larger dataset sizes a less constrained method like standard NNs work well. Generally, though, to also scale to datasets with 100 000 training samples, one can use all the novel long-sequence Transformer variants [1] and create PFNs that scale better. This will be a direction of future work.
>
> **“Isn't $x_i$ one dimensional?”**
>
> No, generally the inputs of examples in each dataset $x_i$ actually have multiple dimensions. Only for Figures 3, 7 and 8 we used a single feature, such that we are able to visualize the PPD. For all other experiments the $x_i$’s in the dataset have many dimensions.
>
> [1] Tay, Y., Dehghani, M., Bahri, D., & Metzler, D. (2020). Efficient transformers: A survey.

---

### Official Review · Reviewer_W5N8 · 2021-11-03

**Correctness:** 3
**Technical Novelty And Significance:** 3
**Empirical Novelty And Significance:** 3
**Recommendation:** 5
**Confidence:** 5

**Main Review:**

Strengths:
1. The proposed framework is clear and sound, and it works well for some well known models (e.g., GP and BNN).

Weaknesses:
1. According to Eq(2) or Algorithm 1, the dataset $D$ can be drawn from a much larger dataset or distribution $p(\mathcal{D})$. In practical, the sampled data size of $D$ should not be computationally reasonable, which means that $D$ may not represent $p(\mathcal{D})$ very well. I'm wondering the scalability of this approach. In other works, for more stochastic case ($|D|$ is far smaller than $\mathcal{D}$ or a much larger model with more number of parameters), how about the performance of the proposed method? In the experiment, only small datasets are discussed.
2. In this paper, the authors emphasize that the transformer encoder can be adapted to fulfill the suggested PPD approximation. So does the title. However, I didn't see the necessity or justification to use transformer as the tool for PPD approximation. By observing Fig 2(a), I think any graph neural networks can achieve the same purpose.

**Summary Of The Paper:**

This paper presents the posterior inference framework named Prior-Data Fitted Networks. The theoretical background is to approximate the posterior predictive distribution (PPD) with a proposal distribution and optimize the KL divergence between the proposal and the PPD. The architecture used as approximation networks is adapting from a transformer encoder, by tweaking the attention mask to output the estimation of $y$ for queries alone. In addition, the authors mainly discussed two application cases, Gaussian process (GP) and Bayesian neural networks (BNN). In the experiment, the paper demonstrates the proposed method can be successful in few-shot learning.

**Summary Of The Review:**

In summary, the paper is clearly written and proposes an interesting method for approximation of posterior predictive distribution. The author has validated their approach in the classification task with small datasets. But I would like to see the generalization on other tasks and data/model scalability. In addition, I have concerns or unclear intuition for adopting transformer encoder architecture.

---

> ### Author Response · Authors · 2021-11-09
> **Reply to Review by Reviewer W5N8**
>
> Dear Reviewer W5N8,
>
> Thank you for reviewing our paper and for your valuable feedback, which we would like to answer in detail now.
>
> **Answer to Weakness 1**
>
> In this paper we sample datasets from a distribution over datasets $D \sim p(\mathcal{D})$ during training, see Figure 1. We sample a lot of these artificial datasets from $p(\mathcal{D})$ (up to 4 million in our experiments, but we could sample as many as needed, with performance continuing to improve with this number and the only constraint being to keep the compute time reasonable). These artificial datasets are used for training our PFNs, by feeding one dataset at a time (as a whole) to the model.
> These datasets are artificial and are not supposed to be the same as those used in evaluation; they are merely used to specify a prior for Bayesian inference. So, while each sampled dataset $D$ might not be representative of the whole distribution $p(\mathcal{D})$, we can sample as many datasets as we need to cover the distribution well enough.
>
> In our experiments, we consider datasets $D$ with up to $|D|=2000$ examples, see Figure 4.
> We believe this is a relevant dataset scale, as Bayesian methods tend to perform the strongest compared to methods based on point estimates, like standard neural network training, for small datasets. For large datasets like CIFAR-10 or ImageNet it is less important to be Bayesian and the strongest models are in fact not Bayesian, but point estimates (see https://paperswithcode.com/sota/image-classification-on-imagenet). For small tabular datasets (Section 6) or few-shot image recognition (Section 7) on the other hand we can show that our Bayesian method outperforms previous methods.
>
> Did this answer your concern? (We were not entirely sure that we understood exactly what was unclear, and if the above does not answer your intended question, could you please clarify?)
>
> **Answer to Weakness 2**
>
> We agree that there are various different architectures that could have been used in principle to address the set-valued input. Indeed, we chose to call our method “prior-fitted networks” rather than “prior-fitted transformers” in order to not rule out future models that work better. Transformers were simply the model most intuitive to us given the constraints. One could have used a graph neural network as well, but wouldn’t this yield a relatively trivial bi-partite graph?
>
>
> Finally, in the summary you mention, that you would like to see applications to more tasks. We would like to point out that we have applied our method to multiple GP priors and BNN priors, each with different dataset sizes, kernels, hyper-parameters and feature dimensions. In addition we provide experiments on image data for the few-shot learning case.
> Do you have a further application in mind that is clearly missing?

---

### Official Review · Reviewer_ZRNF · 2021-11-08

**Correctness:** 3
**Technical Novelty And Significance:** 3
**Empirical Novelty And Significance:** 3
**Recommendation:** 8
**Confidence:** 4

**Main Review:**

Strong points:

- The meta-learning algorithm itself, which proposes to draw datasets from synthetic priors, is an interesting and novel approach. It requires a leap of faith to attempt to transfer knowledge from synthetic datasets, such as ones generated from the BNN, to real-world datasets such as the tabular ones. It is surprising and exciting that this approach worked well and beat all the baselines.

- The novel set-valued training approach which allows to feed entire datasets as samples, and the appropriately modified Transformer model. It fits very well with the meta-learning approach, and allows for fast inference.

- The experiments are convincing and quantitative. In particular, the authors compare their approach to very solid baselines with appropriate tuning.

Weak points:

Mostly, I would like to see a bit more details and explanations, as outlined in the questions and comments below:

Section 2:
Starting with ‘in this work’, this is not background anymore, this is your method. I would rework that into section 3.

Section 4:
How do you choose the size of the datasets, N? Is this a hyperparameter of sorts? Is it better to have datasets of say, 100 samples each, or 10k samples?

In section 5.1, am I correct in my understanding that what you’re doing is:
1- Choosing a given gaussian process.
2- Sampling from the GP.
3- Grouping the samples into buckets of size N: these are the datasets used for training your PFN.
4- Training the PFN on the datasets, and in figure 4)a) you vary the number of datasets you use (from 500K to 4M)
5- Using that for inference on new, unseen datasets generated from the GP (1000 datasets, according to appendix F).

If this is correct, then I don’t understand how this is different from traditional supervised learning, in which you directly train on all the datapoints without grouping into datasets. Indeed, I understand the point of meta-learning as being that the datasets are coming from different data generating processes. Section 5.2 and 5.3 make sense to me, as the data generating process changes between sample datasets due to the sampling of GP parameters.

I don’t understand what does ‘number of data points’ refer to in Figure 4? Is the size of the dataset, N?

Section 5.3:

In figure 5, you show the performance of a PFN with 4M datasets. However, in the text, you mention sampling K=100 BNNs with m=200 and n=100. Are those just the evaluation datasets?

Section 6:

Am I correct in my understanding that, for this section, the data priors for the PFN are just the synthetic datasets generated in sections 5.2 and 5.3 (plus the architecture prior)? As in, there is no PFN training on the tabular datasets, only bayesian inference?

If it is the case, that is very impressive, and I wonder why does it work so well? In particular, why does the scale and nature of features (e.g. categorical) not impact the appropriateness of the data priors you used? Why did you not fine-tune on the tabular datasets, just like you did in section 7 for Omniglot? I’m surprised because seemingly the tasks in the tabular datasets are very different from the GP or BNN priors, since the data generating process is intuitively very different.


**Summary Of The Paper:**

This paper describes a novel way to do bayesian inference with deep learning models by employing a meta-learning approach. This is done by creating a synthetic prior over datasets, training a deep learning model on samples from this prior, which results in an approximation of the posterior predictive distribution over the datasets. The authors demonstrate their approach on two synthetic tasks: approximating gaussian processes and bayesian neural networks, then on two real world tasks: the OpenML tabular datasets and the few-shot learning Omniglot dataset.

**Summary Of The Review:**

Overall, I vote for accepting. I think the approach is very novel and the results surprisingly good – especially for the tabular data with BNN priors. This method has a lot of potential for bridging the gap between deep learning models and small data regimes, all the while adopting a Bayesian perspective. It also opens the door to very interesting future research, for instance on the question of design of priors for a particular problem.
My main concern is with regards to the clarity of the paper, as there is a lot of prior knowledge assumed from the reader and sometimes a lack of details and explanation. Hopefully my comments will be addressed during the rebuttal period.

---

> ### Author Response · Authors · 2021-11-12
> **Reply to Review by Reviewer ZRNF**
>
> Dear Reviewer ZRNF,
>
> Thank you for reviewing our paper and your very useful feedback! You raised valuable concerns, which we would like to answer in detail. We will therefore cite questions from your review and answer them below.
>
> > Section 4: How do you choose the size of the datasets, N? Is this a hyperparameter of sorts? Is it better to have datasets of say, 100 samples each, or 10k samples?
>
> Thank you for bringing this up. To make it clearer here: $N=n+m$ is the sum of the number of training samples $n$ and the number of evaluation samples $m$ in the prior fitting step. In our experiments we sample the number of training samples up to some maximum, see Appendix D.1. The number of evaluation samples m does not have a large effect on the training. It does not need to be representative of the evaluation size in PFN inference, however choosing this to be very small leads to more noisy gradients, as this implicitly decreases the batch size.
>
>
> > In section 5.1, am I correct in my understanding that what you’re doing is: 1- Choosing a given gaussian process. 2- Sampling from the GP. 3- Grouping the samples into buckets of size N: these are the datasets used for training your PFN. 4- Training the PFN on the datasets, and in figure 4)a) you vary the number of datasets you use (from 500K to 4M) 5- Using that for inference on new, unseen datasets generated from the GP (1000 datasets, according to appendix F).
> > If this is correct, then I don’t understand how this is different from traditional supervised learning, in which you directly train on all the datapoints without grouping into datasets. Indeed, I understand the point of meta-learning as being that the datasets are coming from different data generating processes. Section 5.2 and 5.3 make sense to me, as the data generating process changes between sample datasets due to the sampling of GP parameters.
>
> That isn't quite correct. We draw a sample from the GP, or another prior, and then sample N data points from it. So we are: 1. Choosing a given gaussian process. 2. Sampling N data points from the GP 3. Repeating steps 1 & 2 for the number of synthetic training datasets used. 4. Training the PFN on the datasets.
> It is correct that we vary the number of synthetic datasets from 500K to 4M and apply the model on unseen data.
>
> The crucial point here is that when sampling from a GP, each sample will yield a different function. See here for example: https://peterroelants.github.io/posts/gaussian-process-tutorial/#Sampling-from-prior. We show in Theorem 1, that optimizing over datasets generated in this way leads to a PPD approximation. That is, we learn the model of the PPD but not the data itself. In supervised learning one would train on the data of interest X using a gradient-based method, while in our setup we learn the PPD in a gradient-based manner, but predicting on X is replaced by one inference step of the architecture.
>
> > I don’t understand what does ‘number of data points’ refer to in Figure 4? Is the size of the dataset, N?
>
> Thank you for bringing this up. This refers to the small 'n', (N=m+n), i.e. the number of training samples in each dataset. Revealing more training samples leads to better predictions under the true GP posterior as well as our method.
>
> > In figure 5, you show the performance of a PFN with 4M datasets. However, in the text, you mention sampling K=100 BNNs with m=200 and n=100. Are those just the evaluation datasets?
>
> Yes, these are just the evaluation datasets, which we listed for reproducibility. The usage of $K$ was actually a typo here (fixed now, see https://openreview.net/revisions/compare?id=KSugKcbNf9&left=JcOUG-a1Pl&right=c2vZkiMwM7R&pdf=true, thanks; in our notation, $K$ is the number of *training* datasets) . We are using 100 datasets for evaluation, with $m=200$ evaluation data points and $n=100$ revealed data points per dataset. $K=4M$ datasets were used in training the model. Thanks for raising this.

---

> > ### Author Response · Authors · 2021-11-12
> > **Continuation of Reply to Review by Reviewer ZRNF**
> >
> > > Am I correct in my understanding that, for this section, the data priors for the PFN are just the synthetic datasets generated in sections 5.2 and 5.3 (plus the architecture prior)? As in, there is no PFN training on the tabular datasets, only bayesian inference?
> >
> > That is correct, fine tuning was only used in the Omniglot experiment.
> >
> > > If it is the case, that is very impressive, and I wonder why does it work so well? In particular, why does the scale and nature of features (e.g. categorical) not impact the appropriateness of the data priors you used? Why did you not fine-tune on the tabular datasets, just like you did in section 7 for Omniglot? I’m surprised because seemingly the tasks in the tabular datasets are very different from the GP or BNN priors, since the data generating process is intuitively very different.
> >
> > We believe this highlights the importance of priors in models with few training samples given. Neural Networks perform poorly in this regime since solutions are underconstrained. Methods like linear regression and boosted trees provide priors that are methodologically easy to capture but not necessarily valid. However the space of BNNs within the range of architectures in the prior is quite large and captures a large number of possible models.
> > We normalized features (using training samples only) to the same distribution as in the synthetic datasets. If we evaluate our models‘ performance (with a BNN prior) on datasets with many/few categorical features no clear pattern is visible, i.e. models seem to handle categorical features well as it is. That might be since a BNN prior with nonlinear activations is sufficiently complex to handle the nonlinearity of categorical feature effects.
> > We did not fine tune on the tabular datasets to clearly demonstrate the effectiveness of the simple priors. However, it is likely that fine tuning would improve the performance of the approach as well, like for Omniglot.

---

### Public Comment · ~Jakob_H._Macke1 · 2021-11-10
**Description of relationship with simulation-based inference?**

This proposal looks very cool, but it might useful to more clearly explain how the proposal relates to work in the field of simulation-based inference? There has been a lot of work on how neural networks can learn bayesian inference on simulations sampled from priors, see e.g.  https://www.pnas.org/content/117/48/30055 https://arxiv.org/abs/1605.06376 https://arxiv.org/abs/1703.00868 https://arxiv.org/abs/2002.03712 https://proceedings.neurips.cc/paper/2018/file/2e9f978b222a956ba6bdf427efbd9ab3-Paper.pdf as entry points into the literature (beyond these papers, many loss functions, network models and learning approaches have been proposed). Many of these papers are motivated by models in which likelihoods are hard to compute, but of course they can also be used to speed up inference at test-time by amortising inference. It would seem useful to relate this paper to work in this field so that its specific contributions can be communicated more clearly, and that it is appropriately situated in the context of prior work. It is somewhat surprising that none of this work is discussed, and that none of the reviewers asked for it.

---

> ### Author Response · Authors · 2021-11-12
> **Paper Update Proposal to Include Related Work in Simulation-Based Inference**
>
> Dear Jakob Macke,
>
> Thank you for pointing us to this related line of research that we were not aware of. This is a very valuable input to us. We would like to include a paragraph on prior work in simulation-based inference in the paper. What would be your opinion on the following changes to our paper?
>
> **i) An additional paragraph on simulation-based inference in Section 2:**
>
> > Another strain of research related to PFNs is amortized simulation-based inference [0,2]. The goal of amortized simulation-based inference is to find a model that approximates the posterior $p(\theta|X)$ for a prior $p(\theta)$ and a data simulator $p(X|\theta)$ based on training on samples from $p(X,\theta)$. We, on the other hand, approximate the PPD in the supervised learning case, that is, we approximate $p(y|x,\mathcal{D})$ based on samples from $p(\mathcal{D})$.
> Previous work in simulation-based inference is mostly focused on simulations for which a specific model underlying the data is known [1]; we additionally focus on general priors, like GPs and BNNs, that provide a means to tackle fast supervised learning in a more general setting.
>
> [0] Cranmer et al. https://www.pnas.org/content/pnas/117/48/30055.full.pdf
>
> [1] Lueckmann et al. http://proceedings.mlr.press/v130/lueckmann21a/lueckmann21a.pdf
>
> [2] Chan et al. https://proceedings.neurips.cc/paper/2018/file/2e9f978b222a956ba6bdf427efbd9ab3-Paper.pdf
>
> **ii) The last point of our future work can then also be reformulated:**
>
> *Current*
> > (iv) Work on approximating the posterior for parametric models with PFNs, instead of their PPD. The same setup proposed here could in principle be used for that task, too.
>
> *Updated*
> > (iv) Work on using our model for the amortized simulation-based inference setting, where a posterior over latent variables of a parametric model is approximated.
>
> The application of our architecture to this setting could be promising as well.
>
> We would be very glad for your opinion of our assessment.

---

> > ### Public Comment · ~Jakob_H._Macke1 · 2021-11-15
> > **relationship with SBI**
> >
> > Obviously I am just posting here as an interested reader, and its the opinion of the reviewers and AC that is much more pertinent ;-)
> >
> > After thinking about it more:   Neural posterior estimation (NPE) a particular SBI approach is probably of most relevance here: It deals with doing amortised inference on p(theta|x), given a forward model p(theta|x) with parameters theta and a prior p(theta). The approach is to generate samples from the joint p(theta,x), and then to teach a network to learn p(theta|x) my minimising the corresponding log-loss using supervised learning.
> >
> > If I understand it correctly, your proposal focuses on p(y|x), where x is data and y are labels-- the approach is to generate samples from the joint p(x,y), and then to teach a network to learn p(y|x) by minimising the corresponding log-loss using supervised learning.
> >
> > So, I think that there might be overlap between the approach you propose (e.g. the flow chart and possibly the initial lemmata, versions of which have probably come up in different contexts) and neural posterior estimation (e.g. An Le et al, Papamakarios et al, Greenberg et al). But of course, there are very big and important differences: -- in SBI, x is typically a single vector (sometimes a dataset, as e.g. in Chan et al), and theta is multi-dimensional -- hence, much of the work is on the 'output' side of the inference networks (e.g. to express multi-dimensional posterior distributions, often via normalising flows). In contrast, in your approach, the input are data-sets to condition on as training sets and test x,  and the output are single labels-- so much of the work seems to be on the 'input' side of the inference network. So, 'network-architecturally' the two approaches seem to be dealing with complementary challenges. And of course, the empirical challenges and results are likely very different.
> >
> > Btw, I think the Lueckmann paper you referenced above really takes a somewhat different approach, it uses a neural network to learn a forward model, but uses MCMC for inference.

---

> > > ### Author Response · Authors · 2021-11-24
> > > **More Remarks / Revision**
> > >
> > > Thank you very much for your comments, this is a great contribution to us. Regarding your points:
> > > - Lueckmann et al. published multiple papers on SBI; one deals with a method using MCMC for inference, but the paper we cite is a benchmark paper that reviews a number of methods, including (S)NPE.
> > > - Regarding similarities in the objective of the supervised learning function, we do very much agree that the difference is in the realm of a change of variables and data modalities. (Often theta is shared between data points, i.e. a parameter of the model and not an observation.)
> > >
> > > We updated our paper to pay tribute to that fact.

---

### Author Response · Authors · 2021-11-24
**Final Revision of the Discussion Period**

We thank our reviewers and Jakob Macke for their suggestions. We uploaded a final revision with the following changes:
- We added all hyper-parameters for the BNN on PFNs for tabular data.
- We ran the MCMC and SVI baseline for higher budgets for Figure 5, to see where their NLL converges for BNNs.
- We added simulation-based inference to our “Background” section, as well as further related work in meta learning.
- We separated between the explanation of our method and prior work more clearly, by removing explanations of our method from the “Background” section.

Please see the diff for a detailed change log: https://openreview.net/revisions/compare?id=KSugKcbNf9&left=c2vZkiMwM7R&right=KkrrsIFmWp&pdf=true.

---

### Decision · Program_Chairs · 2022-01-20

**Decision:**

Accept (Poster)

**Comment:**

This paper presents a method for using transformer models to perform approximate Bayesian inference, in the sense of approximating the posterior predictive distribution for a test example.  This seems similar to doing amortized variational inference using a transformer model.  The reviewers all found the paper to be clearly written, interesting, novel and compelling.  Two of the reviewers found the results "impressive".  There is some concern of over-claiming (is it really Bayesian?, are the authors making too broad statements based on very simple case studies?).  The presented method is also not scalable O(n^2), so the setting is restricted to very small datasets and models.
 However, the reviewers didn't seem especially concerned by this.  The reviews were mixed but leaning positive (8, 6, 5) and the positive reviews are more substantial.  Therefore the recommendation is to accept, but please incorporate the reviewer feedback and additional discussion about related methods (discussion below) into the camera ready.